# MOLECULAR CONFORMATION GENERATION VIA SHIFTING SCORES

## ABSTRACT

Molecular conformation generation, a critical aspect of computational chemistry, involves producing the three-dimensional conformer geometry for a given molecule. Generating molecular conformation via diffusion requires learning to reverse a noising process. Diffusion on inter-atomic distances instead of conformation preserves SE(3)-equivalence and shows superior performance compared to alternative techniques, whereas related generative modelings are predominantly based upon heuristical assumptions. In response to this, we propose a novel molecular conformation generation approach driven by the observation that the disintegration of a molecule can be viewed as casting increasing force fields to its composing atoms, such that the distribution of the change of inter-atomic distance shifts from Gaussian to Maxwell-Boltzmann distribution. The corresponding generative modeling ensures a feasible inter-atomic distance geometry and exhibits time reversibility. Experimental results on molecular datasets demonstrate the advantages of the proposed shifting distribution compared to the state-of-the-art.

## 1 INTRODUCTION

The molecular conformation generation task constitutes a crucial and enabling aspect of numerous research pursuits, particularly in the study of molecular structures and their potential energy landscapes (Strodel, 2021). Traditional computational methods for this task rely on optimizing the free energy grounded on Schrodinger equation or density functional theory or its approximations (Griffiths & Schroeter, 2018; Tsujishita & Hirono, 1997; Labute, 2010), failing to find a good balance between complexity and quality. Recently, machine learning has emerged as a powerful and efficient tool to identify more stable and diverse conformations across an expanded chemical space (Xu et al., 2021b; Ganea et al., 2021; Xu et al.; Jing et al.). However, such novel approaches give rise to some new challenges.

One of the most significant challenges is incorporating the roto-translational equivariance (SE(3)-equivariance) intrinsic to the generation process. Recent works employ SE(3)-equivariant molecular properties as proxies to render the model invariance. For instance, some studies focus on predicting torsional angles (Jing et al.; Ganea et al., 2021) or inter-atomic distances (Simm & Hernández-Lobato, 2020; Xu et al.; Ganea et al., 2021), with the final conformation assembled through post-processing. Besides, Uni-Mol (Zhou et al., 2023a) predicts the delta coordinate positions based on atom pair representation to update coordinates. Other works leverage inter-atomic distances to directly predict coordinates using generative models (Xu et al.; Shi et al., 2021; Xu et al., 2021b; Zhu et al.). In parallel with these efforts, researchers have developed SE(3)-equivariant graph neural networks (GNNs) to better characterize the geometry and topology of geometric graphs (Schütt et al., 2017; Satorras et al., 2021; Han et al., 2022). These GNNs serve as effective tools or backbones for molecular conformation generation (Jing et al.; Ganea et al., 2021; Xu et al.; Shi et al., 2021; Xu et al., 2021b; Hoogeboom et al., 2022).

Following the previous works (Xu et al.; Shi et al., 2021; Xu et al., 2021b), our approach also seeks to encode SE(3)-equivariance from an inter-atomic distance perspective. To the best of our knowledge, existing works do not yet provide a systemic analysis of distance, often relying on common or heuristic Gaussian assumption on distance changes (Xu et al., 2021b). In this study, we conduct a thorough analysis of inter-atomic distances, drawing inspiration from physical atom motion phenomena. Specifically, we investigate the disintegration process of molecular structures

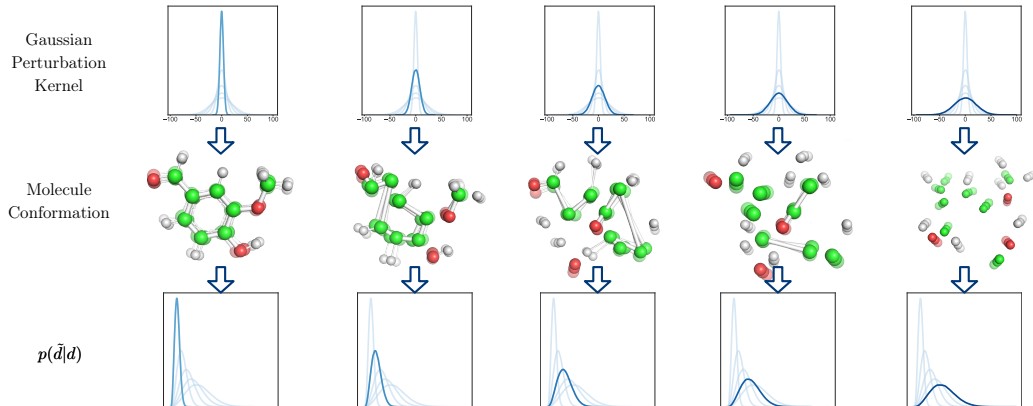

Figure 1: Demostration of the diffusion process of SDDiff. As the Gaussian perturbation level on *atom coordinates* increases, the distribution of *inter-atomic distances* shifts from Gaussian to Maxwell-Boltzmann, which SDDiff learns to reverse.

and aim to learn how to reverse these processes for generating conformations. To this end, the disintegration of molecules can be viewed as being caused by the introduction of gradually increasing levels of perturbing force fields. We postulate that atoms within a molecule exhibit Brownian motion (Gaussian) under relatively small perturbing forces. When the forces are considerably large, chemical structures are disrupted, and the atoms are able to move without restrictions. In this stage, the atom speeds follow a Maxwell-Boltzman distribution. Naturally, this can be connected to the distance distribution, in accordance with the escalation of perturbation intensity. See Fig. 1 for an overview.

We thus put forth a precise estimation of the perturbed distance distribution through a closed-form shifting score function. Further, we propose a novel diffusion-based model named **SDDiff** (shifting distance diffusion) to reverse the force field to recover molecule conformations, leading to superior performance.

Our main contributions are:

- Inspired by molecule thermodynamics, we show that under the Gaussian perturbation kernel on molecular conformation, the distribution of relative speeds and the change of inter-atomic distances shift from Gaussian to Maxwell-Boltzmann distribution.
- We propose a diffusion-based generative model, SDDiff, with a novel and closed-form shifting score kernel, with the mathematical support and empirical verification of its correctness.
- Our method achieves state-of-the-art performance on two molecular conformation generation benchmarks, GEOM-Drugs (Axelrod & Gómez-Bombarelli, 2022) and GEOM-QM9 (Ramakrishnan et al., 2014).

## 2 RELATED WORK

**Molecular conformation generation**. Learning techniques are increasingly equipped for molecular conformation generation. A shallow trial is GeoMol (Ganea et al., 2021), which predicts local 3D configurations and assembles them with heuristic rules. Instead, conformations can be holistically sampled via modelings of either inter-atomic distance (Shi et al., 2021; Simm & Hernández-Lobato, 2020) or atom coordinates (Xu et al.; Zhu et al.). Recently, a rising interest has been observed in diffusion-based approaches (Shi et al., 2021; Xu et al., 2021b; Jing et al.), where the most related works to ours are ConfGF (Shi et al., 2021) and GeoDiff (Xu et al., 2021b). ConfGF perturbs the distance and estimates the corresponding score, which is subsequently converted to the coordinate score via chain rule. However, such a process may result in infeasible 3D geometry. GeoDiff perturbs coordinates instead and introduces an SE(3)-equivariant Markov kernel transiting the coordinate diffusion process to the distance process. However, this model's design is based on the assumption that the perturbed distance follows a Gaussian distribution. This heuristic assumption can lead to mismatches and inaccuracy.

**Diffusion-based generative models**. Denosing diffusion probabilistic models (DDPM) (Ho et al., 2020) delineates a Markov chain of diffusion steps to add random noise to data and subsequently learns to invert the diffusion process for generating desired data samples. Analogous to DDPM, the score matching with Langevin dynamics (SMLD) models (Song & Ermon, 2019; 2020) train noise conditional score networks (NCSN) that approximate the score function of the dataset and apply the stochastic gradient Langevin dynamics to approximate the data distribution. The above two models can be unified under the context of stochastic differential equations (SDEs) (Song et al., 2020b). The denoising diffusion implicit model (DDIM) (Song et al., 2020a) has a controllable sampling stochasticity, allowing the generation of higher-quality samples with fewer steps. The latent diffusion model (LDM) (Rombach et al., 2022) is another accelerated sampler by implementing the diffusion process in the latent space.

$SE(3)$ **Neural Networks**. The Euclidean group, denoted as $SE(3)$ or $E(3)$ when including reflections, represents a group of symmetries in 3D translation and rotation. Due to the geometric symmetry nature of molecules, incorporating this property in feature backbones is essential. One typical line of research is related to GNNs. Schnet (Schütt et al., 2017) is an $E(n)$-invariant network for modeling quantum interactions in molecules. E(n)- Equivariant Graph Neural Networks (EGNNs) (Satorras et al., 2021) is an $E(n)$-equivariant GNN, which does not rely on computationally expensive higher-order representations in intermediate layers. A hierarchy-based GNN named Equivariant Hierarchy-based Graph Networks (EGHNs) (Han et al., 2022) can increase the expressivity of message passing, which is also guaranteed to be $E(3)$-equivariant to meet the physical symmetry. Another related line of research is not restricted to the message-passing paradigm (Gilmer et al., 2017). Some existing works (Thomas et al., 2018; Fuchs et al., 2020) utilize the spherical harmonics to compute a basis for the transformations, which preserve $SE(3)$-equivariance.

## 3 BACKGROUND

### 3.1 MOLECULAR CONFORMATION GENERATION

The generation of molecular conformation can be regarded as a generative problem conditioned on a molecular graph. For a given molecular graph, it is required to draw independent and identically distributed (i.i.d.) samples from the conditional probability distribution $p(\mathcal{C}|G)$, in which $p$ adheres to the underlying Boltzmann distribution (Noé et al., 2019), while $\mathcal{C}$ and $G$ signify the conformation and formula of the molecule, respectively. Formally, each molecule is depicted as an undirected graph $G = (V, E)$, with $V$ representing the set of atoms within the molecule and $E$ denoting the set of inter-atomic chemical bonds, as well as the corresponding node features $\boldsymbol{h}_v \in \mathbb{R}^f, \forall v \in V$ and edge features $\boldsymbol{e}_{uv} \in \mathbb{R}^{f'}, \forall (u, v) \in E$ representing atom types, formal charges, bond types, etc. To simplify the notation, the set of atoms $V$ in 3D Euclidean space is expressed as $\mathcal{C} = [\mathbf{c_1}, \mathbf{c_2}, \cdots, \mathbf{c_n}] \in \mathbb{R}^{n \times 3}$, and the 3D distance between nodes $u$ and $v$ is denoted as $d_{uv} = ||\mathbf{c_u} - \mathbf{c_v}||$. A generative model $p_{\boldsymbol{\theta}}(\mathcal{C}|G)$ is developed to approximate the Boltzmann distribution.

### 3.2 EQUIVARIANCE IN MOLECULAR CONFORMATION

Equivariance under translation and rotation ($SE(3)$ groups) exhibits multidisciplinary relevance in a variety of physical systems, hence plays a central role when modeling and analyzing 3D geometry (Thomas et al., 2018; Weiler et al., 2018; Chmiela et al., 2019; Fuchs et al., 2020; Miller et al., 2020; Simm et al., 2020; Batzner et al., 2022). Mathematically, a model $\mathbf{s}_{\boldsymbol{\theta}}$ is said to be equivariance with respect to $SE(3)$ group if $\mathbf{s}_{\boldsymbol{\theta}}(T_f(\mathbf{x})) = T_g(\mathbf{s}_{\boldsymbol{\theta}}(\mathbf{x}))$ for any transformation $f, g \in SE(3)$. Utilizing conformational representations directly to achieve equivariance presents challenges in accurately capturing the chemical interactions between atoms. Consequently, this approach may result in the generation of molecular structures with inaccuracies and poor configurations. An alternative approach is to use the inter-atomic distance that is naturally equivariant to $SE(3)$ groups (Shi et al., 2021; Xu et al., 2021b; Gasteiger et al., 2020), which will be further introduced in Sec. 4.2.

### 3.3 LEARNING VIA SCORE MATCHING

**Langevin dynamics**. Given a fixed step size $0 < \epsilon \ll 1$, take $\mathbf{x}_0 \sim \pi(\mathbf{x})$ for some prior distribution and use Euler–Maruyama method for simulating the Langevin dynamics

$$\mathbf{x}_t = \mathbf{x}_{t-1} + \frac{\epsilon}{2}\nabla_{\mathbf{x}} \log p(\mathbf{x}_{t-1}) + \sqrt{\epsilon}\mathbf{z}_t, \tag{1}$$

where $\mathbf{z}_t \sim \mathcal{N}(\mathbf{0}, \mathbf{I})$. As $t \to \infty$, $\mathbf{x}_t$ can be considered as a sample draw from $p(\mathbf{x})$ under some regularity conditions (Welling & Teh, 2011). This implies that if we know the score function $\nabla_{\mathbf{x}} \log p(\mathbf{x})$, we can use Langevin dynamics to sample from $p(\mathbf{x})$.

**Denosing score matching**. The process of denoising score matching (Vincent, 2011) involves the perturbation of data $\mathbf{x}$ in accordance with a predetermined perturbing kernel, denoted by $q_\sigma(\tilde{\mathbf{x}}|\mathbf{x})$. The objective $\mathbf{s}_{\boldsymbol{\theta}}$ that minimize the following:

$$\frac{1}{2}\mathbb{E}_{q_\sigma(\tilde{\mathbf{x}}|\mathbf{x})p_{\text{data}}(\mathbf{x})}\left[\|\mathbf{s}_{\boldsymbol{\theta}}(\tilde{\mathbf{x}}) - \nabla_{\tilde{\mathbf{x}}}\log q_\sigma(\tilde{\mathbf{x}} \mid \mathbf{x})\|_2^2\right] \tag{2}$$

satisfies $\mathbf{s}_{\boldsymbol{\theta}^*}(\mathbf{x}) = \nabla_{\mathbf{x}}\log q_\sigma(\mathbf{x})$ almost surely (Vincent, 2011). This implies that to train a denoising model $\mathbf{s}_{\boldsymbol{\theta}}$, we can set the loss functions to be

$$\mathcal{L}\left(\mathbf{s}_{\boldsymbol{\theta}}; \{\sigma_i\}_{i=1}^L\right) \triangleq \frac{1}{L}\sum_{i=1}^L \lambda(\sigma_i)\,\ell(\mathbf{s}_{\boldsymbol{\theta}}; \sigma_i) \tag{3}$$

$$\ell(\mathbf{s}_{\boldsymbol{\theta}}; \sigma) \triangleq \frac{1}{2}\mathbb{E}_{p_{\text{data}}(\mathbf{x})}\mathbb{E}_{\tilde{\mathbf{x}}\sim q_\sigma(\tilde{\mathbf{x}}|\mathbf{x})}\|\mathbf{s}_{\boldsymbol{\theta}}(\tilde{\mathbf{x}}, \sigma) - \nabla_{\tilde{\mathbf{x}}}\log q_\sigma(\tilde{\mathbf{x}}|\mathbf{x})\|_2^2. \tag{4}$$

where $\lambda(\sigma) \propto 1/\mathbb{E}\left[\|\nabla_{\tilde{\mathbf{x}}}\log p_\sigma(\tilde{\mathbf{x}} \mid \mathbf{x})\|_2^2\right]$ is a reweighting coefficient so that the magnitude order of the loss function does not depend on $\sigma$ (Song et al., 2020b). After obtaining a model $\mathbf{s}_{\boldsymbol{\theta}^*}(\mathbf{x}) \approx \nabla_{\mathbf{x}}\log q_\sigma(\mathbf{x})$, following the (annealed) Langevin dynamics (Song & Ermon, 2019), one can draw sample from $p_{\text{data}}(\mathbf{x})$ by recursive computing $\tilde{\mathbf{x}}_t = \tilde{\mathbf{x}}_{t-1} + \frac{\alpha_i}{2}\mathbf{s}_{\boldsymbol{\theta}}(\tilde{\mathbf{x}}_{t-1}, \sigma_i) + \sqrt{\alpha_i}\mathbf{z}_t$, where $\alpha_i = \epsilon \cdot \sigma_i^2/\sigma_L^2$.

**Maxwell-Boltzmann distribution**. In the domain of statistical mechanics, the Maxwell-Boltzmann (MB) distribution serves as a model for delineating the velocities of particles within idealized gaseous systems. These systems are characterized by freely moving particles within a stationary enclosure, where interactions among the entities are negligible apart from momentary collisions. From a mathematical perspective, the MB distribution is the $\chi$-distribution with three degrees of freedom (Young et al., 2008). The probability density function of $\text{MB}(\sigma)$ is given by $f_\sigma(x) = \sqrt{\frac{2}{\pi}}\frac{x^2 e^{-x^2/(2\sigma^2)}}{\sigma^3}$ with support $\mathbb{R}_{++}$.

## 4 METHODOLOGY

### 4.1 MODELING THE DISTRIBUTION OF INTER-ATOMIC DISTANCES

In the present investigation, molecular disintegration is facilitated by the application of progressively intensified perturbation force fields. Upon perturbing a single atom, adjacent atoms experience a consequent force, arising from the chemical bonds interconnecting them with the perturbed atom. In case when a relatively minor perturbative force field is employed, chemical bonds remain unbroken, thereby restricting atomic motions. This observation leads us to hypothesize that individual atoms exhibit Brownian motions under such conditions. Contrarily, when a sufficiently potent force field is imposed, chemical bonds are destroyed, permitting atoms to undergo virtually uninhibited motion with the bare occurrence of collisions. We further hypothesize that the relative speed between any two atoms adheres to the Maxwell-Boltzmann (MB) distribution. Focusing on the inter-atomic distances $d$ within a molecule, we establish that the marginal distribution of perturbed inter-atomic distances $\tilde{d}$, given $d$, is equivalent to the distribution of relative velocities among the atoms.

Specifically, let $\sigma_t$ measure the perturbing force fields at time $t$ and $\{\sigma_t\}_{t=0}^T$ is an increasing non-negative sequence. Then,

$$p_{\sigma_0}(\tilde{d}|d) = p_{\sigma_0}(v) = \mathcal{N}(\tilde{d}|d, 2\sigma_0^2\mathbf{I}), \qquad p_{\sigma_T}(\tilde{d}|d) = p_{\sigma_T}(v) = \text{MB}(\sqrt{2}\sigma_T). \tag{5}$$

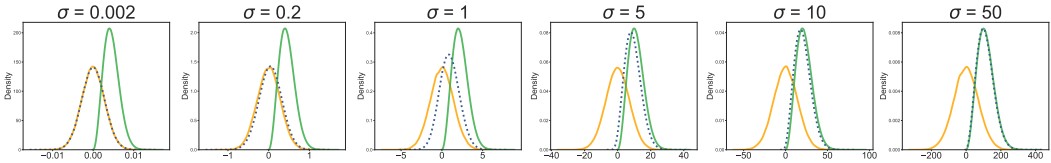

Figure 2: In the investigation of perturbed distance distributions resulting from the introduction of Gaussian noise to molecular conformation, a transition from Gaussian to MB is observed as the noise level escalates. The perturbation's intensity is denoted by $\sigma$. Within the graphical representation, the orange curve delineates the pdf of $\mathcal{N}(0, 2\sigma^2)$, the green curve corresponds to the pdf of $\mathrm{MB}(\sqrt{2}\sigma)$, and the blue dotted curve represents the pdf of $p(\tilde{d}|d)$.

For intermediate perturbing forces, we set $p_{\sigma_t}(\tilde{d}|d) \propto \tilde{d}^{2f_\sigma(\tilde{d},d)} e^{-\frac{(\tilde{d}-d)^2}{4\sigma_t^2}}$, where several constrains are on $f_\sigma$. For a smoothly shifting perturbing force field, we require $f_\sigma(\tilde{d}, d)$ to be smooth with respect to $\sigma, \tilde{d}$ and $d$. To make the limiting perturbing force field be Gaussian and MB, we require $\lim_{\sigma\to0} f_\sigma = 0$ and $\lim_{\sigma\to\infty} f_\sigma = 1$. Thus, we have (note that when $\sigma_T$ is sufficiently large, $\tilde{d} - d \approx \tilde{d}$)

$$p_{\sigma_0}(\tilde{d}|d) \propto e^{-\frac{(\tilde{d}-d)^2}{4\sigma_0^2}} \quad \propto \mathcal{N}(\tilde{d}|d, 2\sigma_0^2 \mathbf{I}) \tag{6a}$$

$$p_{\sigma_T}(\tilde{d}|d) \propto \tilde{d}^2 e^{-\frac{(\tilde{d}-d)^2}{4\sigma_T^2}} \propto \mathrm{MB}(\sqrt{2}\sigma_T) \tag{6b}$$

If we take $f_\sigma(\tilde{d}, d) = 1 - e^{-\sigma/d}$,

$$\nabla_{\tilde{d}} \log q_\sigma(\tilde{d} \mid d) = \left(1 - e^{-\sigma/d}\right) \frac{2}{\tilde{d}} - \frac{\tilde{d} - d}{2\sigma^2} \tag{7}$$

We can simply use a Gaussian kernel as an approximation of perturbing force fields acting on the molecule conformation, i.e., $p_\sigma(\tilde{\mathcal{C}}|\mathcal{C}) = \mathcal{N}(\tilde{\mathcal{C}}|\mathcal{C}, \sigma^2\mathbf{I})$, for $\mathcal{C} \in \mathbb{R}^{n\times3}$, so that the limiting distributions of atoms' speed and conditional perturbed inter-atomic distance are Gaussian and MB distributions. This is because

$$\tilde{\mathcal{C}}_u = \mathcal{C}_u + \mathbf{z}_u \qquad \tilde{\mathcal{C}}_v = \mathcal{C}_v + \mathbf{z}_v \qquad \text{where } \mathbf{z}_u, \mathbf{z}_v \sim \mathcal{N}(\mathbf{0}, \sigma^2\mathbf{I})$$

$$\tilde{d}_{uv} = \|\mathbf{z} + \mathcal{C}_u - \mathcal{C}_v\| \qquad (\mathbf{z} = \mathbf{z}_u - \mathbf{z}_v \sim \mathcal{N}(\mathbf{0}, 2\sigma^2\mathbf{I}))$$

$$= \|\mathcal{C}_u - \mathcal{C}_v\| + \|\mathbf{z} + \mathcal{C}_u - \mathcal{C}_v\| - \|\mathcal{C}_u - \mathcal{C}_v\|$$

$$= d_{uv} + \frac{2\mathbf{z}^\top(\mathcal{C}_u - \mathcal{C}_v) + \|\mathbf{z}\|^2}{\|\mathbf{z} + \mathcal{C}_u - \mathcal{C}_v\| + \|\mathcal{C}_u - \mathcal{C}_v\|}$$

When $\sigma$ is sufficiently small, $\tilde{d}_{uv} \approx d_{uv} + \frac{2\mathbf{z}^\top(\mathcal{C}_u - \mathcal{C}_v)}{2\|\mathcal{C}_u - \mathcal{C}_v\|} = d_{uv} + \hat{z}$, where $\hat{z} \sim \mathcal{N}(0, 2\sigma^2)$. When $\sigma$ is sufficiently large, $\tilde{d}_{uv} \approx d_{uv} + \frac{\|\mathbf{z}\|^2}{\|\mathbf{z} + \mathcal{C}_u - \mathcal{C}_v\|} \approx \|\mathbf{z}\|$, where $\|\mathbf{z}\| \sim \mathrm{MB}(\sqrt{2}\sigma)$. For a comprehensive elucidation of intermediary mathematical procedures, we direct the readers to Appendix A. We conduct experiments to verify the above mathematical derivation. In the conducted experiments, Gaussian perturbations with varying levels of variation are introduced to molecular conformations, i.e., $p(\tilde{\mathcal{C}}|\mathcal{C}) = \mathcal{N}(\mathbf{0}, \sigma^2\mathbf{I})$, for $\mathcal{C} \in \mathbb{R}^{n\times3}$, and the marginal distributions of the difference in inter-atomic distances before and after perturbation are examined. The resultant observations can be seen in Fig. 2 and 3.

## 4.2 MODELING CONFORMATIONS

We model the inter-atom distances instead of the conformation for equivariance as discussed in Sec. 3.2. Consider molecules formed by $n$ atoms, where $n \geq 5$. Given any $C \in \mathbb{R}^{n\times3}/\mathrm{SE}(3)$, let

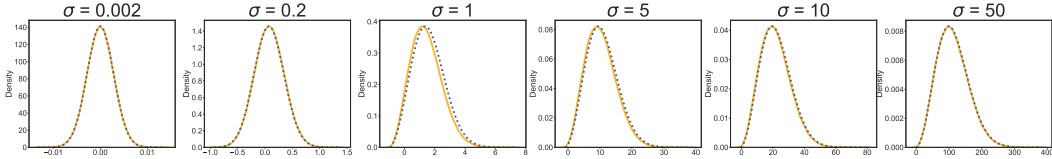

Figure 3: Distribution approximation. The actual pdf $p_\sigma(\tilde{d} - d|d = \text{const})$ is illustrated by the orange curve, whereas the blue dotted curve signifies the proposed approximated pdf.

$d(\cdot) : \mathbb{R}^{n \times 3} / \operatorname{SE}(3) \to \mathbb{D}$ be the mapping from conformations to all inter-atomic distances, where $\mathbb{D} := \operatorname{image}(d)$. Hence, $\mathbb{R}^{n \times 3} / \operatorname{SE}(3)$ and $\mathbb{D}$ are isomorphisms since to ascertain the relative position of a particular point, it is merely necessary to determine its distances from 4 other non-coplanar distinct points. We use $d_{ij}$ to denote the entry $(i, j)$ of the adjacent matrix and we have, by slight abuse of notations

$$\nabla_{\tilde{\mathcal{C}}} \log q_\sigma(\tilde{\mathcal{C}}|\mathcal{C}) = \frac{\partial}{\partial \tilde{\mathcal{C}}} \log q_\sigma(\tilde{\mathcal{C}}, d(\tilde{\mathcal{C}})|\mathcal{C}, d(\mathcal{C})) \tag{8a}$$

$$= \sum_{i,j} \frac{\partial d_{ij}(\tilde{\mathcal{C}})}{\partial \tilde{\mathcal{C}}} \frac{\partial}{\partial d_{ij}(\tilde{\mathcal{C}})} \log q_\sigma(d(\tilde{\mathcal{C}})|d(\mathcal{C})) \quad \text{(almost surely)} \tag{8b}$$

$$= \sum_{i,j} \frac{\partial \tilde{d}_{ij}}{\partial \tilde{\mathcal{C}}} \nabla_{\tilde{d}_{ij}} \log q_\sigma(\tilde{d}|d) \tag{8c}$$

The above property also holds for $\hat{d}(\cdot)$ that maps the conformation to a partial distance vector where each atom is associated with at least 4 distances. A previous work (Shi et al., 2021) showed that for any $\mathbf{s}_\theta(\tilde{d}) \approx \nabla_{\tilde{d}} \log q_\sigma(\tilde{d}|d)$ as a function of the perturbed inter-atomic distance $\tilde{d}$, the scoring network $\mathbf{s}_\theta$ is equivariant w.r.t. $\operatorname{SE}(3)$.

By Eq. 3, 4, 8c and 7, the denoising score matching objective for conformations is

$$\mathcal{L}\left(\boldsymbol{\theta}; \{\sigma_i\}_{i=1}^L\right) \triangleq \frac{1}{L} \sum_{i=1}^L \lambda(\sigma_i) \ell(\boldsymbol{\theta}; \sigma_i) \tag{9a}$$

$$\ell(\boldsymbol{\theta}; \sigma) = \frac{1}{2} \mathbb{E}_{p_{\text{data}}(d)} \mathbb{E}_{p_\sigma(\tilde{d}|d)} \left\| \mathbf{s}_\theta(\tilde{d}, \sigma) - \frac{\partial \tilde{d}}{\partial \tilde{\mathcal{C}}} \left[ \left(1 - e^{-\sigma/d}\right) \frac{2}{\tilde{d}} - \frac{\tilde{d} - d}{2\sigma^2} \right] \right\|_2^2 \tag{9b}$$

Note that $\nabla_{\tilde{\mathcal{C}}} \log q_\sigma(\tilde{\mathcal{C}} \mid \mathcal{C}) \neq -\frac{\tilde{\mathcal{C}} - \mathcal{C}}{\sigma^2}$ since $\tilde{\mathcal{C}}, \mathcal{C} \in \mathbb{R}^{n \times 3} / \operatorname{SE}(3)$ and the probability density function is different from that in $\mathbb{R}^{n \times 3}$. Take $\lambda(\sigma_i) = \sigma_i^2$, $\lambda(\sigma_i) \ell(\boldsymbol{\theta}; \sigma_i) \propto 1$ for any $\sigma_i$. Thus, the loss magnitude order of the loss function does not depend on the specific selection of $\sigma_i$.

### 4.3 NETWORK FOR MODELING CONFORMATION SCORE

The network employed for the purpose of modeling $\mathbf{s}_\theta$ must adhere to two specific criteria which are delineated in Sec. 4.2. For simplification, we omit the model's parameter of molecular graph $\mathcal{G}$.

$\operatorname{SE}(3)$ **equivariance**. It is imperative that the network abstains from utilizing molecular conformation directly as input; rather, it should incorporate inter-atomic distance to achieve $\operatorname{SE}(3)$ equivariance. The employment of perturbed distance as a means to directly forecast the conformation score necessitates a domain transition, thereby augmenting the complexity of the learning process. Thus, following the parametrization of the conformation score as discussed in Sec. 4.2, a generative model for estimating the score of distances is formulated, followed by the application of the chain rule to facilitate the conversion of distance scores into their corresponding values for conformation scores.

**Isomorphisms.** Each individual atom must be associated with a minimum of four distances, in order to establish isomorphisms between $C \in \mathbb{R}^{n \times 3} / \operatorname{SE}(3)$ (representing conformation space) and $\mathbb{D}$ (signifying feasible inter-atomic distance space). On the other hand, correlating an atom with an

excessive number of distances exacerbates the challenge for the model to generate a feasible $d$. The underlying reason for this complication is the disparity in cardinal numbers of $\mathbb{R}^{n \times 3}/\operatorname{SE}(3)$ and $\mathbb{D}$. $\mathbb{D}$ is a subset of $\mathbb{R}^m_{++}$, where $m = \binom{n}{2}$ is the number of edges in complete graph induced by the molecule. For a more detailed illustration, we refer readers to Appendix B. As a result, we connect the three-hop neighborhood in each chemical molecule so that almost every atom in a molecule is connected with at least four other atoms.

Following GeoDiff (Xu et al., 2021b), we adapt a similar network for modeling $\mathbf{s}_\theta$. Given an input graph $G$, the Message Passing Neural Networks (MPNN) (Gilmer et al., 2017) is adopted as $\mathbf{s}_\theta$, which computes node embeddings $\boldsymbol{h}_v^{(t)} \in \mathbb{R}^f, \forall v \in V$ with $T$ layers of iterative message passing:

$$\boldsymbol{h}_u^{(t+1)} = \psi\left(\boldsymbol{h}_u^{(t)}, \sum_{v \in N_u} \boldsymbol{h}_v^{(t)} \cdot \phi(\boldsymbol{e}_{uv}, d_{uv})\right) \tag{10}$$

for each $t \in [0, T-1]$, where $N_u = \{v \in V | (u, v) \in E\}$, while $\psi$ and $\phi$ are neural networks, e.g. implemented using multilayer perceptrons (MLPs). Note that the node features, distances and edge features are input into $\mathbf{s}_\theta$ as initial embeddings when $t = 0$, but we only keep the distance $d$ in the above sections as the input of $\mathbf{s}_\theta$ for notation simplification. Besides, as no coordinates information is explicitly engaged in this network, this kind of modeling can preserve the above two properties. For more details about this part, refer to Appendix B.

## 4.4 Sampling by Langevin dynamics

The learned score matching network $\mathbf{s}_\theta$ that minimizes Eq. 9a can approximate the score of molecular conformation and following the annealed Langevin dynamics, we provide the pseudocode of the sampling process in Alg. 1 from which we can draw conformations given molecule.

---

**Algorithm 1** Sampling via annealed Langevin dynamics

---

**Input**: molecular graph $G$, network $\mathbf{s}_\theta$, scheduler $\{\sigma_i\}_{i=1}^T$.
**Output**: conformation $\mathcal{C}$.
1: Sample $\mathcal{C} \sim \mathcal{N}(\mathbf{0}, \sigma_T^2 \mathbf{I})$.
2: **for** $i = T, T-1, \cdots, 1$ **do**
3: $\quad \alpha_i \leftarrow \epsilon \cdot \sigma_i^2/\sigma_T^2$ $\qquad\qquad$ {$\alpha_i$ is the step size.}
4: $\quad$ Sample $\mathbf{z_i} \sim \mathcal{N}(\mathbf{0}, \mathbf{I})$
5: $\quad \mathcal{C}_{i-1} \leftarrow \mathcal{C}_i + \alpha_i \mathbf{s}_\theta(d(\mathcal{C}_i), \sigma_i) + \sqrt{2\alpha_i}\mathbf{z_i}$ {Langevin dynamics.}
6: **end for**
7: **return** $\mathcal{C}_0$

---

## 4.5 Analysis

**Marginal v.s. joint distributions**. From existing literature, the diffusion models are built on adding isotropic Gaussian noise $\mathcal{N}(\mathbf{0}, \sigma^2\mathbf{I})$ to the modeled objects such as pixel values in image generations. In SDDiff, we add isotropic Gaussian noise to molecule conformation (coordinate), and noise is mapped to inter-atomic distances. Thus, entries of noise on distance are not independent, whereas the marginal distribution of distances can be applied for score matching, this is because

$$\nabla_{\tilde{d}_i} \log p_\sigma(\tilde{d} \mid d) = \nabla_{\tilde{d}_i} \log p_\sigma(\tilde{d}_i | d_{1,2,\cdots,m}) \cdot p_\sigma(\tilde{d}_{1,2,\cdots,i-1,i+1,\cdots,m} \mid d_{1,2,\cdots,m}, \tilde{d}_i, d_i)$$
$$= \nabla_{\tilde{d}_i} \log p_\sigma(\tilde{d}_i | d_i) + \nabla_{\tilde{d}_i} \log p_\sigma(\tilde{d}_{N(i)} \mid d_{N(i)}, \tilde{d}_i, d_i) \approx \nabla_{\tilde{d}_i} \log p_\sigma(\tilde{d}_i | d_i)$$

where $N(i)$ is the set of edge indices whose edges are incident with edge $i$. The second equality holds because $\tilde{d}_i$ gives no information on the distribution of other perturbed edges that are not incident with edge $i$. Also, $d_j$ gives no information on the distribution of $\tilde{d}_i$ where $i \neq j$. We hypothesize that disregarding the term $\nabla_{\tilde{d}_i} \log p_\sigma(\tilde{d}_{N(i)} \mid d_{N(i)}, \tilde{d}_i, d_i)$ introduces no bias. This supposition stems from the observation that possessing knowledge of both $\tilde{d}_i$ and $d_i$, we remain uninformed about the increase or decrease in the value of $\tilde{d}_{N(i)} - d_{N(i)}$.

**Approximation by optimal transportation (OT).** Given the knowledge of the distributions at end time points $p_{t=0}(x)$ and $p_{t=T}(x)$, the problem of obtaining the distributions in between can be formulated as a Shrodinger Bridge problem whose solution is also the solution of entropic OT. We compute the regularized Wasserstein Barycenter of $p_{t=0}(\tilde{d}|d)$ and $p_{t=T}(\tilde{d}|d)$ by employing the approach presented in a previous work (Benamou et al., 2015). However, the regularization term

Table 1: Results of molecular conformation generation.

| Methods | GEOM-QM9 | | | | GEOM-Drugs | | | |
| | COV(%) ↑ | | MAT(Å) ↓ | | COV(%) ↑ | | MAT(Å) ↓ | |
| | Mean | Median | Mean | Median | Mean | Median | Mean | Median |
|---|---|---|---|---|---|---|---|---|
| CGCF | 78.05 | 82.48 | 0.4219 | 0.3900 | 53.96 | 57.06 | 1.2487 | 1.2247 |
| ConfVAE | 77.84 | 88.20 | 0.4154 | 0.3739 | 55.20 | 59.43 | 1.2380 | 1.1417 |
| GeoMol | 71.26 | 72.00 | 0.3731 | 0.3731 | 67.16 | 71.71 | 1.0875 | 1.0586 |
| ConfGF | 88.49 | 94.31 | 0.2673 | 0.2685 | 62.15 | 70.93 | 1.1629 | 1.1596 |
| GeoDiff | 90.54 | 94.61 | 0.2090 | 0.1988 | 89.13 | 97.88 | 0.8629 | 0.8529 |
| **SDDiff (ours)** | **91.07** | **94.69** | **0.2048** | **0.1941** | **90.68** | **98.48** | **0.8564** | **0.8503** |

impacts the limiting weighted Barycenter, leading to divergences from $p_{t=0}(\tilde{d}|d)$ to $p_{t=T}(\tilde{d}|d)$. As a result, the regularized Wasserstein Barycenter approach is unsuitable for intermediate distribution approximation. See Appendix C for a more detailed analysis.

## 5 EXPERIMENT

### 5.1 EXPERIMENT SETTINGS

**Datasets**. We use two widely used datasets, GEOM-QM9 (Ramakrishnan et al., 2014) and GEOM-Drugs (Axelrod & Gómez-Bombarelli, 2022) for evaluating molecular conformation generation. The GEOM-QM9 dataset comprises molecules with an average of 11 atoms, while the GEOM-Drugs dataset consists of larger molecules with an average of 44 atoms. For a fair comparison, we adopted the same dataset split as GeoDiff (Xu et al., 2021b). For both datasets, the training set contains 40k molecules, the validation set contains 5k molecules and the test set contains 200 molecules. Please refer to GeoDiff (Xu et al., 2021b) for more details regarding the dataset.

**Evaluation metrics**. We use the metrics of COV (coverage) and MAT (matching) (Xu et al.) to measure both diversity and accuracy. Specifically, we align ground truth and generated molecules by the Kabsch algorithm (Kabsch, 1976), and then calculate their difference with root-mean-square-deviation (RMSD). Then the COV and the MAT are defined as follows:

$$\text{COV} = \frac{1}{|S_r|}\{\mathcal{C} \in S_r | \text{RMSD}(\mathcal{C}, \mathcal{C}') < \delta, \exists \mathcal{C}' \in S_g\}, \quad \text{MAT} = \frac{1}{|S_r|}\sum_{\mathcal{C}' \in S_g} \text{RMSD}(\mathcal{C}, \mathcal{C}')$$

where $S_g$ and $S_r$ denote generated and ground truth conformations, respectively. Following some baselines (Xu et al., 2021b; Ganea et al., 2021), we set the threshold of COV $\delta = 0.5$ Å for GEOM-QM9 and $\delta = 1.25$ Å for GEOM-Drugs, and generate twice the number of ground truth conformation for evaluation.

**Baselines**. We choose 5 state-of-the-art models for comparison: GeoMol (Ganea et al., 2021) is not a generative model that generates conformation by hand with predicted molecular information. CGCF (Shi et al., 2021) is a two-step method, and ConfVAE (Xu et al., 2021a) is a VAE-based model. ConfGF (Shi et al., 2021) and GeoDiff (Xu et al., 2021b) are two similar works that are also diffusion-based.

Other implementation details are provided in Appendix D

### 5.2 RESULTS AND ANALYSIS

The results of molecular conformation generation are shown in Table 1. The baseline results are obtained from GeoDiff (Xu et al., 2021b). In order to mitigate the impact of the model's backbone and primarily evaluate the efficacy of distance distribution modeling, we have opted to utilize a backbone that closely resembles that of GeoDiff. This will enable us to more accurately assess the performance of the distance distribution modeling technique while minimizing the potential confounding effects

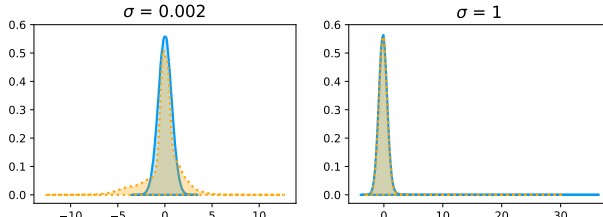

Figure 4: The ground truth depicted in blue is the distribution of $\sigma \nabla_{\tilde{d}} \log p(\tilde{d}|d)$, whereas the distribution of the model's outputs is represented by a dashed orange line. It can be observed that as the value of $\sigma$ increases, $\sigma \nabla_{\tilde{d}} \log p(\tilde{d}|d)$ tends to exhibit the characteristics of a long-tailed Gaussian distribution. For a detailed introduction to the figure, we refer readers to Appendix E.

of the model's underlying architecture. The Visualization of selected generated conformation can be found in Appendix G.

**Score distribution.** In the existing literature, the ground truth score function follows a normal distribution. Specifically, the ground truth of score matching objects is set to $\sigma \nabla_{\tilde{\mathbf{x}}} \log p(\tilde{\mathbf{x}}|\mathbf{x}) \sim \mathcal{N}(\mathbf{0}, \mathbf{I})$. The proposed distance distribution diverges from the Gaussian distribution when the perturbation level is significantly large and requires the model to parametrize a non-Gaussian distribution. In order to investigate the efficacy of existing backbones in approximating such distribution, we visually depict the *distribution of score functions* (not inter-atomic distance), along with our backbone's output under varying levels of perturbation. The ensuing results have been found in Fig. 4. It is evident that our proposed distribution closely resembles the Gaussian distribution when $\sigma$ is reasonably small. Conversely, when $\sigma$ is substantially large, the proposed score function transforms into a long-tailed Gaussian distribution. Despite this alteration, the model's output distribution still approximates the proposed score function effectively. This substantiates that the proposed distribution can be effortlessly approximated, and thus can be incorporated into a wide array of models.

**Planar structure generation** As mentioned in Eq. 8b, the score function of distance can be transformed into the score function of conformation *almost surely*, provided that the conformation is non-planar. Nonetheless, certain molecular structures like benzene rings, exhibit a planar conformation within local regions, which may render this transformation inapplicable (see Fig. 5). A viable solution to optimize these local planar structures further involves utilizing post-processing with variants of rule-based methods (e.g., force field) which encode the unvarying property of certain local structures like benzene rings being planar.

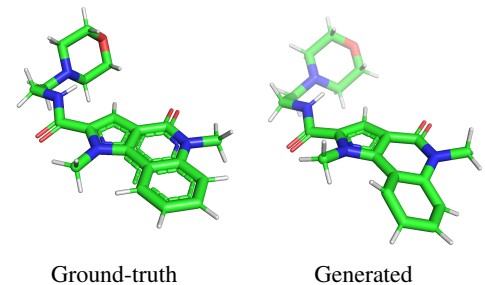

Ground-truth          Generated

Figure 5: Atoms in a benzene ring should be coplanar as ground truth structure, while the generative structure may conflict with such property.

## 6 CONCLUSION

In this study, we present a novel molecular conformation generation approach - SDDiff - by incorporating the shifting score function inspired by molecule thermodynamics. Our main findings include that the distribution of the change of inter-atomic distances shifts from Gaussian to Maxwell-Boltzmann distribution under the Gaussian perturbation kernel on molecular conformation, which can be accurately approximated by our approach. By proposing a diffusion-based generative model with a shifting score kernel, we have provided both the mathematical derivation and experimental validation of its correctness. The effectiveness of our approach has been demonstrated through achieving new state-of-the-art results on two widely used molecular conformation generation benchmarks, namely GEOM-Drugs, and GEOM-QM9. Our method effectively captures the essential aspects of molecular dynamics and inter-atomic interactions, leading to improved performance in generating accurate and feasible molecular conformations.

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

# A   INTERMEDIARY PROCEDURES FOR THE LIMITING DISTRIBUTION OF $p(\tilde{d}|d)$

Given the forward diffusion process of the conformation $\tilde{C} = C + \mathbf{z} \in \mathbb{R}^{n \times 3}$, where $\mathbf{z} \sim \mathcal{N}(\mathbf{0}, \sigma^2 \mathbf{I})$, we have, by slightly abuse of notations

$$\tilde{C}_u = C_u + \mathbf{z}_u \qquad \tilde{C}_v = C_v + \mathbf{z}_v$$

$$\begin{aligned}
\tilde{d}_{uv} &= \|\tilde{C}_u - \tilde{C}_v\| \\
&= \|\tilde{C}_u - C_u + C_u - C_v + C_v - \tilde{C}_v\| \\
&= \|\mathbf{z}_u + \mathbf{z}_v + C_u - C_n\| \\
&= \|\mathbf{z} + C_u - C_v\| \qquad (\mathbf{z} \sim \mathcal{N}(\mathbf{0}, 2\sigma^2 \mathbf{I})) \\
&= \|C_u - C_v\| + \|\mathbf{z} + C_u - C_v\| - \|C_u - C_v\| \\
&= \|C_u - C_v\| + (\|\mathbf{z} + C_u - C_v\| - \|C_u - C_v\|)\frac{\|\mathbf{z} + C_u - C_n\| + \|C_u - C_v\|}{\|\mathbf{z} + C_u - C_v\| + \|C_u - C_v\|} \\
&= \|C_u - C_v\| + \frac{\|C_u - C_v\|^2 + 2\mathbf{z}^\top(C_u - C_v) + \|\mathbf{z}\|^2 - \|C_u - C_v\|^2}{\|\mathbf{z} + C_u - C_v\| + \|C_u - C_v\|} \\
&= \|C_u - C_v\| + \frac{2\mathbf{z}^\top(C_u - C_v) + \|\mathbf{z}\|^2}{\|\mathbf{z} + C_u - C_v\| + \|C_u - C_n\|} \\
&= d_{uv} + \frac{2\mathbf{z}^\top(C_u - C_v) + \|\mathbf{z}\|^2}{\|\mathbf{z} + C_u - C_v\| + \|C_u - C_v\|}
\end{aligned}$$

Take $\sigma \to 0$,

$$\begin{aligned}
\tilde{d}_{uv} &= d_{uv} + \frac{2\mathbf{z}^\top(C_u - C_v) + \|\mathbf{z}\|^2}{\|\mathbf{z} + C_u - C_v\| + \|C_u - C_v\|} \\
&\approx d_{uv} + \frac{2\mathbf{z}^\top(C_u - C_v)}{2\|C_u - C_v\|} \\
&\approx d_{uv} + \mathbf{z}^\top\frac{(C_u - C_v)}{\|C_u - C_v\|} \\
&= d_{uv} + [1, 0, 0]^\top \mathbf{z} \qquad (\mathbf{z} \text{ and } \frac{C_u - C_v}{\|C_u - C_v\|} \text{ are independent}) \\
&= d_{uv} + z_1 \\
\tilde{d}_{uv} &\sim \mathcal{N}(d_{uv}, 2\sigma^2) \quad \text{when } \sigma \text{ is small enough}
\end{aligned}$$

Take $\sigma \to \infty$,

$$\begin{aligned}
\tilde{d}_{uv} &= d_{uv} + \frac{2(\frac{\mathbf{z}}{\|\mathbf{z}\|})^\top(C_u - C_v) + \|\mathbf{z}\|}{\frac{\|\mathbf{z} + C_u - C_u\|}{\|\mathbf{z}\|} + \frac{\|C_u - C_v\|}{\|\mathbf{z}\|}} \\
&\approx d_{uv} + \frac{\|\mathbf{z}\|}{1 + 0} \\
&\approx \|\mathbf{z}\| \\
\tilde{d}_{uv} &\sim \text{MB}(\sqrt{2}\sigma) \quad \text{when } \sigma \text{ is large enough}
\end{aligned}$$

# B   NETWORK FOR MODELING $\mathbf{s}_\theta$

In this section, we introduce the details of the network for modeling $\mathbf{s}_\theta$. In practice, two typical GNNs are adopted to parameterize $\mathbf{s}_\theta$, namely the SchNet (Schütt et al., 2017) and GIN (Xu et al., 2018). SchNet is a deep learning architecture for modeling the quantum interactions in molecules, which is widely used in molecular-related tasks. We adopt SchNet as a global model to generate informative

molecular representations, thus promoting the conditional generation process. Specifically, the SchNet can be formulated as follows:

$$h_u^{(t+1)} = \sum_{v \in N_u} h_v^{(t)} \odot \phi(\exp(-\gamma(e_{uv} - \mu))) \tag{11}$$

where $\phi$ denotes an MLP, $\mu$ and $\gamma$ are two hyperparameters representing the number of Gaussians and the reference of single-atom properties, respectively.

Moreover, as molecules naturally have graph structures, we additionally adopt GIN as a local model, which captures the local structural features. Specifically, the GIN can be formulated as follows:

$$h_u^{(t+1)} = \phi\left((1 + \epsilon) \cdot h_u^{(t)} + \sum_{v \in N_u} \text{ReLU}(h_v^{(t)} + e_{uv})\right) \tag{12}$$

where $\epsilon$ is a trainable parameter and $\phi$ denotes an MLP.

**Isomorphisms.** Each individual atom must be associated with a minimum of four distances, in order to establish isomorphisms between $C \in \mathbb{R}^{n \times 3} / \text{SE}(3)$ (representing conformation space) and $\mathbb{D}$ (signifying feasible inter-atomic distance space). On the other hand, correlating an atom with an excessive number of distances exacerbates the challenge for the model to generate a feasible $d$. The underlying reason for this complication is the disparity in cardinal numbers of $\mathbb{R}^{n \times 3} / \text{SE}(3)$ and $\mathbb{D}$. $\mathbb{D}$ is a subset of $\mathbb{R}_{++}^m$, where $m = \binom{n}{2}$ is the number of edges in complete graph induced by the molecule. When $n = 5, \mathbb{R}^{n \times 3} / \text{SE}(3) \cong \mathbb{D} \cong \mathbb{R}_{++}^m$, since for each $d \in \mathbb{R}_{++}^m$, it corresponds to a conformation $\mathcal{C} \in \mathbb{R}^{n \times 3} / \text{SE}(3)$ and vise versa. When $n > 5$, there exists some infeasible $d \in \mathbb{R}_{++}^m$ such that it corresponds to no conformation in 3D. This is because the inter-atomic distance between a specific atom and its fifth neighbors is uniquely determined by inter-atomic distances between itself and its first four neighbors. As the number of correlated distances of an atom increases, the cardinal number gap between $\mathbb{R}^{n \times 3} / \text{SE}(3)$ and $\mathbb{D}$ increases. Consequently, the absence of a constraint to maintain the model's output within the feasible distance score set (i.e., ensuring that the predicted distances remain feasible upon adding the forecasted distance score to the perturbed score) may result in a violation of the feasibility of the estimated distance. This issue is further exacerbated by the expansion of the number of distances associated with an atom. Identifying a constraint on the model's output that is both differentiable and incurs minimal computational cost can be challenging. As such, an alternative approach is adopted whereby each atom is associated with the least possible number of distances while maintaining a minimum threshold of no fewer than four associated distances. Hence, we connect the three-hop neighborhood of each node.

## C   APPROXIMATING INTERMEDIATE DISTRIBUTIONS VIA ENTROPIC OPTIMAL TRANSPORT

We compute the barycenters of $p_{\sigma=0.1}(\tilde{d} - d)$ and $p_{\sigma=1}(\tilde{d} - d)$, while in SDDiff, $\sigma$ varies in the range of 1e-7 to 12, which results in a more dramatic distribution shift. Following (Benamou et al., 2015), we discretize the empirical distribution of $\tilde{d} - d$ into 800 bins and apply the Sinkhorn algorithm to find barycenters. When the regularization coefficient is set to values less than 5e-4, the Sinkhorn algorithm returns infeasible solutions, i.e., nans. We set the regularization coefficient to 1e-3, 8e-4, and 6e-4. We visualize barycenters computed by the algorithm. The results can be seen in Fig. 6. In the figure, $\alpha$ denotes the weight of two distributions. We see that the approximation accuracy increases as the regularization coefficient $\lambda$ decreases. But with a smaller regularization coefficient, the algorithm collapses. This implies that using the Sinkhorn algorithm to approximate the inter-media distance distribution is unsuitable.

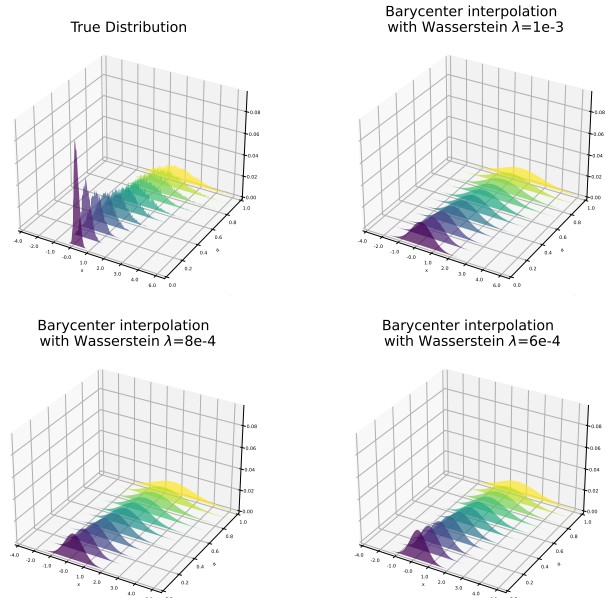

Figure 6: A figure showing the Barycenters under different weights computed by the Sinkhorn algorithm.

## D  IMPLEMENTATION DETAILS

Regarding the implementation details of our experiment, we trained the model on a single Tesla A100 GPU and Intel(R) Xeon(R) Gold 5218 CPU @ 2.30GHz CPU. The training process lasted for approximately 1-2 days, while the sampling took around 8 hours for GEOM-Drugs and 4 hours for GEOM-QM9, respectively. The learning rate was set to 0.001, and we employed the plateau scheduler, which reduced the learning rate to 0.6 every 500 iterations. The optimizer used was Adam, and the batch size was set to 32 for GEOM-Drugs and 64 for GEOM-QM9. We trained the model until convergence, with a maximum of 3 million iterations.

Regarding the diffusion setting, we set the number of steps to 5000 and define $\beta_t$ using a sigmoid schedule ranging from 1e-7 to 2e-3. Then we define $\bar{\alpha}_t = \prod_i^t (1 - \beta_i)$, and $\sigma_t = \sqrt{\frac{\bar{\alpha}_t}{1 - \bar{\alpha}_t}}$. The range of $\sigma_t$ was approximately 0 to 12.

## E  DETAILS OF FIGURE 4

In previous literature, the Gaussian perturbation kernel is employed. Specifically, $\tilde{\mathbf{x}} = \mathbf{x} + \sigma\mathbf{z}$, where $\mathbf{z} \sim \mathcal{N}(\mathbf{0}, \mathbf{I})$, thus $\nabla_{\tilde{\mathbf{x}}} \log p(\tilde{\mathbf{x}}|\mathbf{x}) = \frac{\tilde{\mathbf{x}}-\mathbf{x}}{\sigma^2} = \frac{\mathbf{z}}{\sigma}$. This implies that $\sigma\nabla_{\tilde{\mathbf{x}}} \log p(\tilde{\mathbf{x}}|\mathbf{x}) \sim \mathcal{N}(\mathbf{0}, \mathbf{I})$. The score-matching object is to approximate $\sigma\nabla_{\tilde{\mathbf{x}}} \log p(\tilde{\mathbf{x}}|\mathbf{x})$ and hence model's outputs follow a normal distribution, which exhibits a stable numerical magnitude property. On the contrary, our proposed distribution has that

$$\sigma\nabla_{\tilde{d}_{ij}} \log p(\tilde{d}_{ij}|d_{ij}) = (1 - e^{-\sigma/d_{ij}})\frac{2\sigma}{\tilde{d}_{ij}} - \frac{\tilde{d}_{ij} - d_{ij}}{2\sigma}$$

We visualize the distribution of the above and it has been observed that it follows a long-tailed Gaussian distribution. While the majority of the data points exhibit a Gaussian-like behavior, there exists a low probability of occurrence of samples with significantly large values of $\tilde{d}_{ij} - d_{ij}$. In the event that such cases arise, our backbone architecture is capable of effectively approximating the score function.

## F    COMPARATIVE ANALYSIS OF SDDIFF AND TORSIONAL DIFFUSION

In Sec. 5.2, we have excluded the consideration of Torsional Diffusion (TD) due to significant differences in the modeling paradigms between TD and our proposed SDDiff methodology. In this section, we undertake a comparative discussion of TD and SDDiff to elucidate the distinct characteristics of these approaches.

Firstly, it is important to acknowledge that TD and SDDiff operate within distinct modeling manifolds. SDDiff primarily aims to investigate and address issues existing in models based on inter-atomic distances. The distance-based models need to predict conformers in a manifold of on average 130 dimensions. Differently, TD is based on the torsional angles, significantly reducing the prediction manifold dimension. Statistical analyses of TD (Jing et al.) reveal that it only needs to predict rotatable bonds in a manifold ranging from 3 to 7 dimensions when applied on the Drugs dataset. Thus, TD offers a more manageable learning task, which can lead to improved performance outcomes.

Second, it is worth noting that SDDiff can be purely data-driven, leveraging solely the inherent information present within the molecular dataset. Conversely, TD can process in torsion space to circumvent the challenge of SE(3) invariance relying on ready-made local structures. Before applying diffusion on torsion angles, TD utilizes the RDkit tool to generate stable local structures, which introduces a substantial degree of prior knowledge. RDKit, being a robust chemical toolkit, encodes valuable chemical insights, such as the fixed structure of the benzene ring and hybrid orbital characteristics, which can contribute to the enhancement of prediction quality. The existing work (Zhou et al., 2023b) also proved that combining RDKit and naive clustering techniques could achieve commendable performance for molecular conformation generation. For SDDiff, introducing RDkit before the diffusion process would cast instability to bond lengths while RDKit is necessary to TD. For reference, we adopt a similar strategy to GeoDiff (Xu et al., 2021b) and use Force Field (FF) optimization as post-processing to introduce chemical priors. Table 2 shows the performance improvements achieved through barely FF optimization for both SDDiff and GeoDiff. Also, there still exists room for innovation in enhancing the injection of priors into distance-based models.

Table 2: The results of GEOM-Drugs with FF optimization

| Methods | COV(%) ↑ | | MAT(Å) ↓ | |
| | Mean | Median | Mean | Median |
| --- | --- | --- | --- | --- |
| SDDiff | 90.68 | 98.48 | 0.8564 | 0.8503 |
| SDDiff+FF | 93.07 | 98.18 | 0.7465 | 0.7312 |
| GeoDiff | 89.13 | 97.88 | 0.8629 | 0.8529 |
| GeoDiff+FF | 92.27 | 100.00 | 0.7618 | 0.7340 |

In summary, SDDiff and TD exhibit distinct design philosophies and motivations, rendering a direct comparison between these two models less pertinent. Moreover, the incorporation of chemical priors further underscores the dissimilarities between them. The experimental results in 1 have proven the effectiveness of our proposed distance modeling, and this modeling can serve as a foundational element for the development of models that apply a diffusion process on distances.

## G    GENERATED MOLECULAR CONFORMATION VISUALIZATOIN

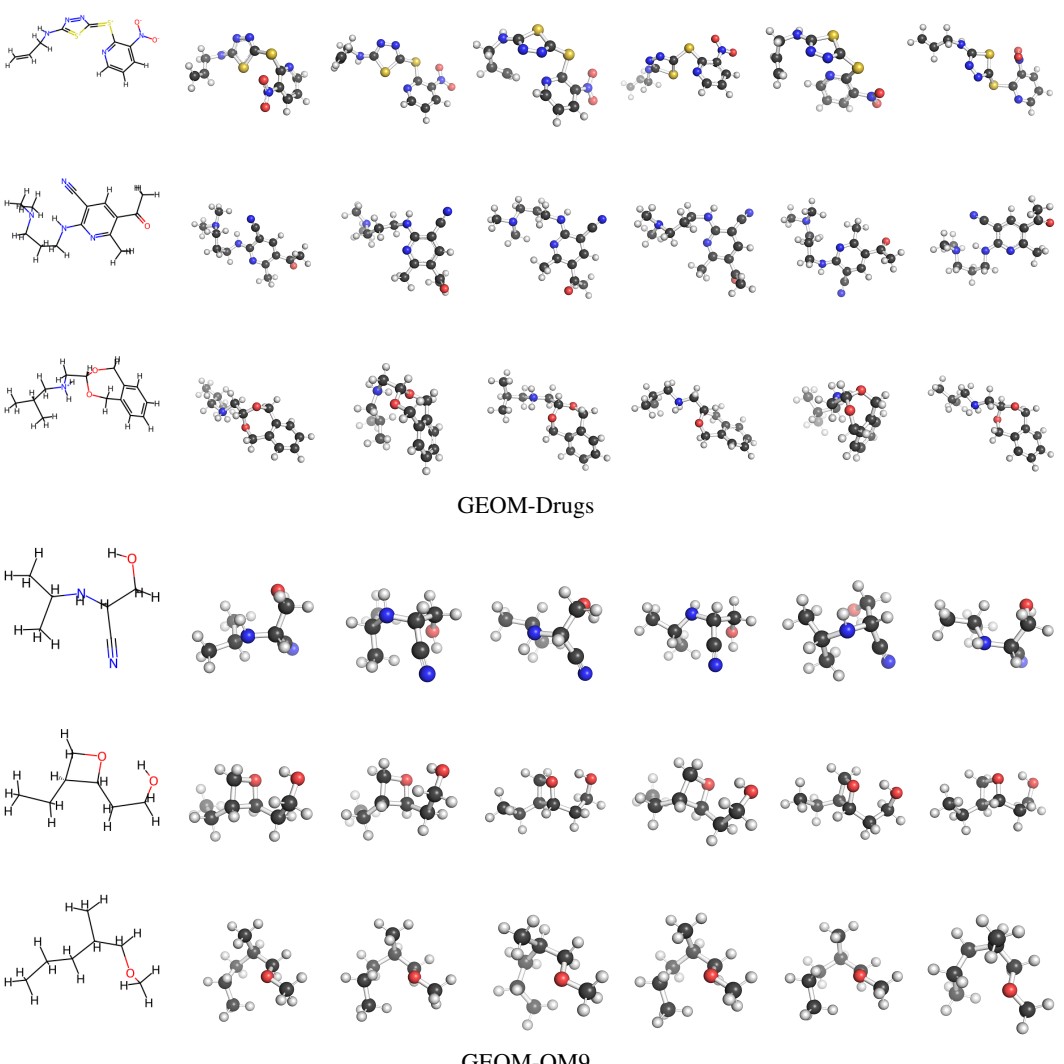

GEOM-Drugs

GEOM-QM9

Figure 7: Selected samples generated with our proposed SDDiff.

