# OpenReview forum: "Molecular Conformation Generation via Shifting Scores"
_ICLR.cc/2024/Conference — Submitted to ICLR 2024_

### Official Review · Reviewer_XVGd · 2023-10-16

**Soundness:** 2 fair
**Presentation:** 1 poor
**Contribution:** 2 fair
**Rating:** 3
**Confidence:** 4

**Summary:**

**Summary**: The authors propose performing diffusion on interatomic distances where the true distances are modeled as a gaussian with small variance and the base distribution is a Maxwell-Boltzmann (MB) distribution with very high variance. They claim this method, SDDiff, preserves SE(3)-equivariance and achieves state-of-the-art results on two molecular conformation benchmarks.

**Strengths:**

**Pros**:

- Diffusion map between gaussian and MB distribution has not been done to the best of my understanding.

- The mathematical derivations in the main text seems sound. I did not check the appendix. Theoretical results are supported with simulations.

**Weaknesses:**

**Cons**:

- *Mischaracterization of prior works*. I do not understand the benefit of diffusion on interatomic distances. The authors claim GeoDiff assumes distances follows a gaussian distribution. Having read GeoDiff, I do not see this assumption. GeoDiff applies diffusion directly on atomic positions. More so, GeoDiff achieves SE(3) equivariance.

- *Missing baseline*. Torsional diffusion has been out for more than a year and achieves state-of-the-art results on the baselines considered in this work. There is no mention of this paper in the related works and it is missing from the baselines. This is a major red flag.

- *Weird theoretical assumption*. The shifting score from gaussian to MB relies on a huge std (\sigma=50) until the perturbation kernel matches MB. First, this seems computationally awkward to have to go distances greater than 400 (figure 2). It seems the huge sigma is avoided with the commonly used scaling trick to control the scale of the score matching objective. However, why is it necessary to go this far and force the base distribution to be MB? Why can't we have a normal diffusion between two gaussians? \sigma=1 looks similar to gaussian to me.

- *Unconvincing results*. The authors claim a new state-of-the-art results. However, they leave out a important baseline and the improvements are extremely small in Table 1. The benefits of SDDiff compared to GeoDiff are within noise. The improvement is at most 0.08 on metrics... Furthermore more difficult benchmarks have been released since [2] almost a year ago. Evaluating against GeoDiff when GeoDiff already achieves 95% seems like the wrong problem to be focusing on.

- *Unexplained analysis*. Section 4.5 is confusing to me. The first part regarding the marginal vs. joint seems to be saying the dependence between distances can be thrown out. This is done without explanation other than a hypothesis and throwing this out makes diffusion on distances no different than diffusion on particles to me. Even in the introduction, diffusion on distances is motivated through the dependence on interatomic forces so throwing them out seems to go against the original motivation. Furthermore, the approximation to OT is very brief and I did not understand the point here.

[1] https://arxiv.org/abs/2203.02923
[2] https://arxiv.org/abs/2206.01729

While I think diffusion on distances rather than particles is interesting for ML on molecules, the formulation in this work confuses me of why it is beneficial (if at all). The results are not convincing and prior works are either mischaracterized or left out. Due these issues, I recomend reject.

**Questions:**

- Why are there negative distances for the gaussian pdf in figure 2?

- The SDDiff authors claim their method is useful in achieving SE(3) equivariance but why is this novel if GeoDiff and related works can already do so?

- How is p_t sampled? This and the training procedure are not specified. What is the \sigma schedule?

---

> ### Author Response · Authors · 2023-11-19
>
> **Weakness**
>
> > 1. Mischaracterization of prior works.
>
> In GeoDiff, authors stated that under the common assumption that $d$ also follows a Gaussian distribution (p7 Chain-rule approach before Sec 4.4). GeoDiff injects the noise on the coordinates and then transforms the distribution of the coordinates into the distribution of the distance. By such a transformation, GeoDiff attains the SE(3) equivalence. We suggest reviewer go through the official code of GeoDiff to have a better understanding of GeoDiff's implementation.
>
> > 2. Missing baseline.
>
> We strongly suggest reviewers go through the **whole** paper more carefully. We mentioned Torsional diffusion in Introduction (paragraph 2, line 3-4) and Related work (line 6). Even in Appendix F, **we use a full page to discuss our work and Torsional diffusion**. Our modeling method is quite different from Torsional diffusion. Our model is purely data-driven while Torsional diffusion uses strong chemical prior. We think that chemical prior can greatly enhance the model's performance. In [1], the authors claim that using the RDkit and a simple clustering algorithm can surpass most of the deep learning models. Hence, we think that including Torsional diffusion is not fair and did not include modeling methods that use strong prior in the comparison.
>
> > 3. Weird theoretical assumption
>
> We plot the case when $\sigma=50$ as an extreme example to show that when $\sigma$ is sufficiently large, the distribution indeed trends to the MB distribution. In practice, the maximum $\sigma$ is chosen to be round $12$. We can use the Figure 2 when $\sigma=10$ as an illustration. In this case, the distribution is still close to the MB distribution.
>
> > 4. Unconvincing results.
>
> The main contribution of this work is to propose a new shifting distribution for a more accurate approximation of the distance distribution, we did not greatly change the modeling method (i.e., model architecture, modeling manifold) for a fair comparison. Hence, the improvement may not seem to be very large. About the benchmarks released in [2], it is essentially the same as the benchmarks in our paper but with a different error tolerance. We set a toy Drugs dataset that only contains 1/10 of the original training set and the same test set as SDDiff. We trained 2 models whose distance distribution hypotheses are Gaussian and the shifting distribution, respectively. The results can be seen in the following table:
>    |                       | GEOM-QM9 |        |         |        | GEOM-Drugs |        |         |        |
> |:---------------------:|:--------:|:------:|:-------:|:------:|:----------:|:------:|:-------:|:------:|
> |         Method        |  COV(%)↑ |        | MAT(Å)↓ |        |   COV(%)↑  |        | MAT(Å)↓ |        |
> |                       |   Mean   | Median |   Mean  | Median |    Mean    | Median |   Mean  | Median |
> | Gaussian distribution |   85.51  |  89.30 |  0.2495 | 0.2378 |    86.02   |  94.24 |  0.9017 | 0.8963 |
> | shifting distribution |   86.86  |  90.60 |  0.2397 | 0.2365 |    88.36   |  97.06 |  0.8940 | 0.8887 |
>
> > 5. Unexplained analysis
>
> To clarify, in this work, **we do not apply the diffusion on distances**. We apply diffusion on coordinates, model the distribution of distances, and then transform this distribution back to the distribution of the coordinates. Since it is extremely difficult to model the dependence of the distances, hence, we hypothesis that this dependence has a limited infect on the final results. In the introduction, our motivation is that by applying diffusion on coordinates, the relative ``speed'' of atoms can be considered as the MB distribution which is the distribution of the speed of particles when particles move freely with the bare occurrence of collisions.
>
> > 6. While I think diffusion on distances rather than particles is interesting for ML on molecules, the formulation in this work confuses me of why it is beneficial (if at all). The results are not convincing and prior works are either mischaracterized or left out. Due these issues, I recomend reject.
>
> **We believe that reviewer XVGd has mischaracterized the work of both GeoDIff and ours.** We went through the official code of the GeoDiff and we are sure that GeoDiff indeed applies the assumption that the distribution of the distance follows the Gaussian distribution. Also, in this work, we did not apply the diffusion to distances. As indicated by unnumbered equations in p5, we directly inject noise on the coordinates and then model the distribution of the distances, which is similar to GeoDiff's implementation. However, we design a more accurate approximation of the distance distribution, which we believe can enhance the model's performance. The results in the ablation studies show that when the dataset is small, such a more accurate approximation of the distribution can greatly enhance the performance.
>
> [1]https://arxiv.org/pdf/2302.07061.pdf
> [2]https://arxiv.org/abs/2206.01729

---

> > ### Author Response · Authors · 2023-11-19
> >
> > **Quesions**
> >
> > > 1. Why are there negative distances for the gaussian pdf in figure 2?
> >
> > The x-axis in Figure 2 is the values of $\tilde{d} - d$, which can be negative. We will clarify this misleading in our final version.
> >
> > > 2. The SDDiff authors claim their method is useful in achieving SE(3) equivariance but why is this novel if GeoDiff and related works can already do so?
> >
> > The main contribution of this paper is that, while previous work achieved SE(3)-invariance by transforming the conformation distribution to distance distribution, they mainly rely on heuristic assumptions such as the distribution of the distance being Gaussian. In this work, we more carefully model the distribution of distances and prove that by using the proposed distribution, we can enhance the model's performance.
> >
> > > 3. How is p\_t sampled? This and the training procedure are not specified. What is the $\sigma$ schedule?
> >
> > We stated the loss function in Eq. 9. We can train a model $s_{\boldsymbol{\theta}}$ which minimizes the Eq. 9 and sample data following Algorithm 1. The training procedure and the sampling procedure have no difference compared with the standard diffusion model, which is also similar to the diffusion introduced in GeoDiff. The novelty of this work is that the proposed diffusion distribution is not studied in previous literature. $\sigma$ schedule is a hyperparameter that controls how much noise is to be injected in the diffusion forward process at time $t$. Our selection of the $\sigma$ scheduler can be found in Appendix D. The maximum value of $\sigma$ is 12.

---

> > > ### Comment · Reviewer_XVGd · 2023-11-20
> > >
> > > I have read the author's comments and other reviews.
> > >
> > > > We strongly suggest reviewers go through the whole paper more carefully. We mentioned Torsional diffusion in Introduction (paragraph 2, line 3-4) and Related work (line 6). Even in Appendix F, we use a full page to discuss our work and Torsional diffusion. Our modeling method is quite different from Torsional diffusion. Our model is purely data-driven while Torsional diffusion uses strong chemical prior. We think that chemical prior can greatly enhance the model's performance. In [1], the authors claim that using the RDkit and a simple clustering algorithm can surpass most of the deep learning models. Hence, we think that including Torsional diffusion is not fair and did not include modeling methods that use strong prior in the comparison.
> > >
> > > I do not see a mention of Appendix F in the main text. As stated in the ICLR Call for papers, "Authors may use as many pages of appendices (after the bibliography) as they wish, **but reviewers are not required to read the appendix**."
> > > There are two places where torsional diffusion is **cited but this is not equivalent to discussing the comparison**. As mentioned, Appendix F is missable if one only reads the main text. The abstract claims state-of-the-art when torsional diffusion is not mentioned. Therefore this is still a missing comparison both quantitatively and comparatively.
> > >
> > > It is still not well motivated why TD cannot be compared to. The task setting allows for utiliizing the local structure. As mentioned by other reviewers, the authors need to experiment on a different task if their goal is to demonstrate why the method is useful. Otherwise TD needs to be discussed in the main text and compared to.
> > >
> > > > The main contribution of this work is to propose a new shifting distribution for a more accurate approximation of the distance distribution, we did not greatly change the modeling method (i.e., model architecture, modeling manifold) for a fair comparison. Hence, the improvement may not seem to be very large. **About the benchmarks released in [2], it is essentially the same as the benchmarks in our paper but with a different error tolerance.**
> > >
> > > I find this unconvincing to not compare with the TD benchmarks.
> > >
> > > > We trained 2 models whose distance distribution hypotheses are Gaussian and the shifting distribution, respectively. The results can be seen in the following table:
> > >
> > > Again the difference seems very marginal.
> > >
> > > There is potentially a interesting methodological and theoretical contribution with the analysis of interatomic distances during diffusion. However, I still believe the presentation is misleading compared to state-of-the-art on molecular conformation. As other reviewers have stated, the experiments need more convincing to show the usefulness. I recommend looking into toy experiments to elucidate the theory and alternative applications if the gains are marginal on molecular conformations. I will not be changing my score for these reasons.

---

> > > > ### Author Response · Authors · 2023-11-21
> > > >
> > > > > About no mention of Torsional Diffusion in paper but in Appendix}
> > > >
> > > > We will put more discussion about Torsional Diffusion and refer readers to the Appendix in our final version.
> > > >
> > > > > About comparison with TD
> > > >
> > > > We replied it in our response just below your comment above.
> > > >
> > > > > About marginal improvement of ablation study
> > > >
> > > > In the ablation study, we can see that our distance approximation can enhance the model compared with the Gaussian hypothesis. However, since the score functions of the Gaussian distribution and our proposed distribution are similar, these two models have similar performance but our proposed distribution indeed improve the performance.

---

### Official Review · Reviewer_1N22 · 2023-10-31

**Soundness:** 3 good
**Presentation:** 3 good
**Contribution:** 2 fair
**Rating:** 5
**Confidence:** 4

**Summary:**

The paper proposes a new method for molecular conformation generation. The main contribution of the paper is that instead of the Gaussian diffusion process, the paper proposed to use transition kernels changing from Gaussians to Maxwell-Boltzmann. This is in correspondence with adding Gaussian noises to molecular structures. The paper shows good mathematical justification for the closed-form score kernel. Experiments also demonstrate the effectiveness of common benchmarks.

**Strengths:**

Originality is good but not surprising. The model follows the existing geometric diffusion models but with novel transition kernels, and the paper well explains the mathematical foundation of the diffusion process.

Quality and clarity are good. The paper is well-presented and easy to follow. The technical details are clearly explained.

**Weaknesses:**

The main weakness from my perspective is the significance of empirical comparison. The improvement over GeoDiff is not significant to me. Could the author provide more ablation study about the $f_\sigma$ function in Eq7, which can help to verify the importance of the proposed MB diffusion distribution.

**Questions:**

I may miss some details, but feel a little confused about defining the diffusion process on the distances. The motivation is "Gaussian on coordinates results in MB distribution on distances". Then, why not add noise on coordinates which can also enable the MB diffusion on distances? I feel the direct perturbation in distances will also potentially result in infeasible geometry?

---

> ### Author Response · Authors · 2023-11-19
>
> **Weaknesses**
>
> > The main weakness from my perspective is the significance of empirical comparison. The improvement over GeoDiff is not significant to me. Could the author provide more ablation study about the function in Eq7, which can help to verify the importance of the proposed MB diffusion distribution.
>
> SDDiff outperforms GeoDiff greatly when the dataset is small. We set a toy Drugs dataset that only contains 1/10 of the original training set and the same test set as SDDiff. We trained 2 models whose distance distribution hypotheses are Gaussian and the shifting distribution, respectively. The results can be seen in the following table:
> |                       |  GEOM-QM9  |        |         |        | GEOM-Drugs |        |         |        |
> |:---------------------:|:----------:|:------:|:-------:|:------:|:----------:|:------:|:-------:|:------:|
> |         Method        | COV(%)↑    |        | MAT(Å)↓ |        |   COV(%)↑  |        | MAT(Å)↓ |        |
> |                       |    Mean    | Median |   Mean  | Median |    Mean    | Median |   Mean  | Median |
> | Gaussian distribution |    85.51   |  89.30 |  0.2495 | 0.2378 |    86.02   |  94.24 |  0.9017 | 0.8963 |
> | shifting distribution |    86.86   |  90.60 |  0.2397 | 0.2365 |    88.36   |  97.06 |  0.8940 | 0.8887 |
>
> **Questions**
> > I may miss some details, but feel a little confused about defining the diffusion process on the distances. The motivation is "Gaussian on coordinates results in MB distribution on distances". Then, why not add noise on coordinates which can also enable the MB diffusion on distances? I feel the direct perturbation in distances will also potentially result in infeasible geometry?
>
> When the noise is injected into the coordinates, if the noise is small, then the distances follow a Gaussian distribution as proposed by GeoDiff. However, if the noise is large enough, the distribution of the distances follows the MB distribution. That's why we proposed a shifting distance distribution between Gaussian and MB distribution for a more accurate approximation of the exact distribution. Directly injecting noise on the distance results in infeasible geometry. This method was proposed in ConfGF, which was mentioned in related works. From the experimental results, SDDiff and GeoDiff are better than ConfGF.

---

### Official Review · Reviewer_QEij · 2023-11-01

**Soundness:** 2 fair
**Presentation:** 3 good
**Contribution:** 3 good
**Rating:** 5
**Confidence:** 4

**Summary:**

The authors present a diffusion model for generating molecular conformations achieving slightly better scores than SOTA. The authors ensure SE(3)-equivariance by using atom-atom distance matrices from which they construct the conformations. The model (SDDiff) uses a novel shifting score loss, that shifts the distribution of interatomic distances between Gaussian and Maxwell-Boltzmann distribution. The proper physical motivation for this remains unclear.

**Strengths:**

The empirical performance is compelling

**Weaknesses:**

-	Probably my biggest problem with the paper is the motivation for using the Maxwell- Boltzmann distribution (MBD). The MBD describes the distribution of the length of velocity vectors in an ideal gas, and to some good approximation even in a real molecule. The authors generate molecular conformations based on distances, velocities are never generated nor used (e.g., the math on page 5 after eq 7 only includes distances). It is also unclear how interatomic distances (which are by definition purely positive) or the length of velocities could ever follow a Gaussian distribution. Therefore, the appearance and meaning of velocity v in eq 5 needs to be explained clearly.
-	Regarding Section 4.5, if a molecule were to break apart, there will be some correlation of atom-atom distances, i.e., if an atom's distance to another one which is far away in the graph increases this will almost certainly also mean that the distance of the neighboring atom to the far away atom increases. This is more correlation than causation, but it does contain information. Additionally, instead hypothesizing can the authors at least show empirically that their statement is true?
-	Regarding the measures COV and MAT. If we assume that a molecule has, e.g., 3 major conformers which are all very close in RMSD. Wouldn’t a model that samples always only one conformer achieve a COV of 1, even though it has never generated the other 2? Also, the definition of MAT seems odd, is there sum over S_r and maybe a min() missing?
-	On page 9: The authors write that it is evident that the proposed distribution matches the Gaussian closely, however, in Fog. 4 the orange distribution appears to be tri-modal Can the authors compute the overlap of orange and blue for the two values of sigma?
- 	In Section 4.2, even though D is defined as image(d), R^(n×3)/SE(3) is not isomorphic to D: a molecule and its mirror image would have the same distance matrices, but they are not the same element of R^(n×3)/SE(3) if they are chiral. In this regard, it would be valuable for the authors to clarify how the proposed method handles the generation of conformers for stereoisomers or enantiomers.
-In Section 5.1, the COV and MAT metrics introduced correspond to the “Recall” version (COV-R and MAT-R, without the typo mentioned below). Some of the baseline methods under comparison, such as GeoDiff (Xu et al., 2022) and Torsional Diffusion (Jing et al., 2022), also include the “Precision” version (COV-P and MAT-P) to assess the quality of the generated conformers. The authors should include the “Precision” metrics in Table 1 as well.

**Questions:**

-	Especially in the introduction several papers are missing the year of publication, e.g., Xu et al., Jing et al, Zhu et al.,  it is therefore not clear which paper is being cited and if multiple occurrences denote the same paper.
-	What do the authors mean by marginal distribution of interatomic distances? If the full set of 3N(3N-1)/2 number of distances are included, this distribution would be even higher dimensional than the 3N-dimensional Boltzmann distribution of Cartesian coordinates.
-	It is not clear how equation 7 follows from 6 nor how it justifies to “simply” use a Gaussian kernel.
-	On the bottom of page 5, where the authors state that n has to be greater-equal to 5. It would be good to mention there that “Each individual atom must be associated with a minimum of four distances, in order to establish isomorphisms between …”
-	The implications of the “Note…” after eq 9b remain unclear.
-	The paragraph on optimal transport, says that the authors use the regularized Wasserstein barycenter but then continue saying that it is not suited. So what exactly do the authors do?
-	The caption of Table 1 is too short. The caption should at least explain the meaning of COV and MAT and refer the reader to their definition in the text.

-	Why exactly is planarity a problem? Don’t 4 neighbors define any point in 3d exactly?

	In Section 5.1, The MAT(-R) metric should be corrected as follows:
MAT=1/|S_r |  ∑_(C∈S_r)▒min┬(C^'∈S_g )⁡RMSD(C,C^' ) .

---

> ### Author Response · Authors · 2023-11-19
>
> **Weakness**
>
> > 1. Probably my biggest problem with the paper is the motivation for using the Maxwell- Boltzmann distribution (MBD). The MBD describes the distribution of the length of velocity vectors in an ideal gas, and to some good approximation even in a real molecule. The authors generate molecular conformations based on distances, velocities are never generated nor used (e.g., the math on page 5 after eq 7 only includes distances). It is also unclear how interatomic distances (which are by definition purely positive) or the length of velocities could ever follow a Gaussian distribution. Therefore, the appearance and meaning of velocity v in eq 5 needs to be explained clearly.
>
> We are not claiming that the distances between atoms follow the Gaussian distribution, we are proving that under small perturbations, the change of the distances, i.e. $\tilde{d} - d$ follows the Gaussian distribution. Also, $d$ nor $\tilde{d}$ is relative to the concept of speed. However, $\tilde{d} - d$ is relative to the speed since it measures the change of the distances.
>
> > 2. Regarding Section 4.5, if a molecule were to break apart, there will be some correlation of atom-atom distances, i.e., if an atom's distance to another one which is far away in the graph increases this will almost certainly also mean that the distance of the neighboring atom to the far away atom increases. This is more correlation than causation, but it does contain information. Additionally, instead of hypothesizing can the authors at least show empirically that their statement is true?
>
> We generate samples $\tilde{d}_1, \tilde{d}_2, d_1, d_2$, where $d_1, d_2$ are the edge lengths of two incident edges and $\tilde{d}_1, \tilde{d}_2$ are edge lengths after perturbation on coordinates. We compute the Pearson r correlation between random variables $\tilde{d}_1 - d_1$ and $\tilde{d}_2 - d_2$. The results show that under different levels of $\sigma_t$, the correlation coefficient is always round 0.24. Hence, we ignore such a correlation.
>
> > 3. Regarding the measures COV and MAT. If we assume that a molecule has, e.g., 3 major conformers which are all very close in RMSD. Wouldn’t a model that samples always only one conformer achieve a COV of 1, even though it has never generated the other 2? Also, the definition of MAT seems odd, is there sum over S\_r and maybe a min() missing?
>
> The COV and MAT measures can cause this problem. We notice that COV-P and MAT-P can reflect the quality of all generated conformers, we will include these metrics later. And thanks for pointing out the wrong definition of MAT in our paper. It should be $\operatorname{MAT-R} =\frac{1}{|S_r|} \sum_{\mathcal{C} \in S_r} \min_{\mathcal{C}' \in S_g} \operatorname{RMSD} (\mathcal{C}, \mathcal{C}')$. we have revised it.
>
> > 4. On page 9: The authors write that it is evident that the proposed distribution matches the Gaussian closely, however, in Fog. 4 the orange distribution appears to be tri-modal Can the authors compute the overlap of orange and blue for the two values of sigma?
>
> The orange distribution is not tri-modal, it is unimodal, and the distribution is very close to Gaussian.
>
> > 5. "In Section 4.2, even though D is defined as image(d), R\^(n×3)/SE(3) is not isomorphic to D: a molecule and its mirror image would have the same distance matrices, but they are not the same element of R\^(n×3)/SE(3) if they are chiral. In this regard, it would be valuable for the authors to clarify how the proposed method handles the generation of conformers for stereoisomers or enantiomers."
>
> When we say the distance matrices, we are referring to the adjacent matrix, i.e., we consider a fully connected graph. Although two chiral molecules have the same set of bound lengths, their adjacent matrices are different. Hence, $D$ is isomorphic to $\mathbb{R}^{n \times 3} / \operatorname{SE(3)}$ if we define $D$ to be the manifold of adjacent matrices (not the manifold of bond length).
>
> > 6. In Section 5.1, the COV and MAT metrics introduced correspond to the “Recall” version (COV-R and MAT-R, without the typo mentioned below). Some of the baseline methods under comparison, such as GeoDiff (Xu et al., 2022) and Torsional Diffusion (Jing et al., 2022), also include the “Precision” version (COV-P and MAT-P) to assess the quality of the generated conformers. The authors should include the “Precision” metrics in Table 1 as well.
>
> Thanks for your suggestion. we will include these metrics later.

---

> ### Author Response · Authors · 2023-11-19
>
> **Questions**
> > 1. Especially in the introduction several papers are missing the year of publication, e.g., Xu et al., Jing et al, Zhu et al., it is therefore not clear which paper is being cited and if multiple occurrences denote the same paper.}
>
> Thanks for your pointing out it. We have revised the references.
>
> > 2.What do the authors mean by marginal distribution of interatomic distances? If the full set of 3N(3N-1)/2 number of distances are included, this distribution would be even higher dimensional than the 3N-dimensional Boltzmann distribution of Cartesian coordinates.
>
> We refer $p(d_{ij})$ as the marginal distribution of interatomic distances where $d_{ij}$ is the distance among the $i$-th node and the $j$-th node. Indeed such modeling would increase the modeling manifold but still this is the so fared best choice to avoid directly dealing with the SE(3)-invariance in Cartesian coordinates.
>
> > 3. It is not clear how equation 7 follows from 6 nor how it justifies to “simply” use a Gaussian kernel.
>
> Eq.6 gives constraints that the distribution of perturbations on distance is proportional to the normal distribution when $\sigma=\sigma_0$, and proportional to MB distribution when $\sigma=\sigma_T$ We construct $f_\sigma(\tilde{d}, d)=1-e^{-\sigma / d}$, then substitute it into $p_{\sigma_t}(\tilde{d} \mid d) \propto \tilde{d}^{2 f_\sigma(\tilde{d}, d)} e^{-\frac{(\tilde{d}-d)^2}{4 \sigma_t^2}}$ and then we can obtain the score function shown as Eq.7 (the $\nabla_{\tilde{d}} \log q_\sigma(\tilde{d} \mid d)$ should be $\nabla_{\tilde{d}} \log p_\sigma(\tilde{d} \mid d)$, we have fixed it).
>
> The justification to "simply" use a Gaussian kernel is detailed after Eq 7 in those unnumbered equations. We perturb the coordinates with Gaussian noise, then the conditional perturbed inter-atomic distance are Gaussian and MB distributions.
>
> > 4. On the bottom of page 5, where the authors state that n has to be greater-equal to 5. It would be good to mention there that “Each individual atom must be associated with a minimum of four distances, in order to establish isomorphisms between …”}
>
> Thanks for your suggestions, We are open to revisiting and refining the wording here in accordance with your suggestions.
>
> > 5. The implications of the “Note…” after eq 9b remain unclear.
>
> Here, we are trying to remind readers that the score function of conformation is not simply $-\frac{\tilde{\mathcal{C}} - \mathcal{C}}{\sigma^2}$. This is because the considered probability density function has the SE(3)-equivalence. Hence, for example, if $\tilde{\mathcal{C}} = \mathcal{C} + t$, where $t$ is a scaler indicating that we move the atoms in $\mathcal{C}$ along x, y, z axis by $t$. In the conditional states of $p(\tilde{\mathcal{C}} \mid \mathcal{C})$, $\tilde{\mathcal{C}}$ and $\mathcal{C}$ represents the same state since we are considering in a SE(3)-invariant space. Hence, the score function of $p(\tilde{\mathcal{C}} \mid \mathcal{C})$ cannot be easily computed. That's why we transform the score function of distances to the score function of conformations.
>
> > 6. The paragraph on optimal transport, says that the authors use the regularized Wasserstein barycenter but then continue saying that it is not suited. So what exactly do the authors do?
>
> We are approximately the distribution between the Gaussian distribution and the MB distribution. Hence, one may consider using the OT to solve the intermediate distribution. However, we discussed that using this method has numerical stability issues. Hence, we are using the proposed methods in our paper to estimate the intermediate distribution. Discussion about OT is to save time for those trying to solve the intermediate distribution by OT method.
>
> > 7. The caption of Table 1 is too short. The caption should at least explain the meaning of COV and MAT and refer the reader to their definition in the text.}
>
> Thank you for your suggestions. We have expended it.
>
> > 8. Why exactly is planarity a problem? Don’t 4 neighbors define any point in 3d exactly?
>
> Consider the case where 4 points are all in the same plane. You may consider there are 4 points in an A4 paper. Then, given the distances between them and the other additional points, you cannot decide whether this point is on the top of the A4 paper or the downside of the A4 paper.
>
> > 9. In Section 5.1, The MAT(-R) metric should be corrected as follows:
>     MAT=1/|S\_r | ∑\_(C∈S\_r)min┬(C\^'∈S\_g )⁡RMSD(C,C\^' ) .}
>
> Thank you so much for pointing it. We have made corrections.

---

### Official Review · Reviewer_K1M9 · 2023-11-01

**Soundness:** 3 good
**Presentation:** 2 fair
**Contribution:** 1 poor
**Rating:** 3
**Confidence:** 3

**Summary:**

The authors point out an interesting connection between the gaussian perturbation kernel and Maxwell-Boltzmann distribution, and propose a diffusion model to learn such shifting score kernels for conformer generation. They perform a standard benchmark and show that their proposed methods have superior performance under the standard metric.

**Strengths:**

The connection between gaussian perturbation and inter-atomic distance shifts are quite interesting, and the authors are able to leverage such observation to learn a diffusion model to learn such shifting scores. It gives an interesting likelihood model on top of many diffusion-based conformer generation models.

**Weaknesses:**

While the observation is interesting and it's great that the authors are able to demonstrate its superior performance, the GEOM benchmark has been used for quite some time now and probably over-optimized, so it's difficult to argue true superiority marginal gain on one benchmark alone. In addition, the majority of the mathematical framework for score matching involving langevin dynamics are not new to this problem either.

**Questions:**

I am happy to re-evaluate my rating if the authors can provide more compelling evidence that the proposed methods are not just another conformer generation model. For instance, showing superior downstream application impact would be very helpful.

---

> ### Author Response · Authors · 2023-11-19
>
> **Weakness**
> > While the observation is interesting and it's great that the authors are able to demonstrate its superior performance, the GEOM benchmark has been used for quite some time now and probably over-optimized, so it's difficult to argue true superiority marginal gain on one benchmark alone. In addition, the majority of the mathematical framework for score matching involving langevin dynamics are not new to this problem either.
>
> In this work, we are not mainly aiming to propose a new baseline model in the task of conformation generation task. We are considering how to model the distance distribution if we apply the diffusion process to coordinates. Such modeling does not only contribute to molecular generation tasks, but can also be applied to any generation task that takes SE(3)-invariance as an important property. We use the GEOM benchmark mainly because many previous works model the distribution of distance such as ConfGF, and GeoDiff. Hence, we compare these models of the distance distribution and demonstrate our distribution approximation is better compared with theirs. For a fair comparison, since other models apply the Langevin dynamics for generation, we still apply the same sampling method.
>
> **Questions**
> > I am happy to re-evaluate my rating if the authors can provide more compelling evidence that the proposed methods are not just another conformer generation model. For instance, showing superior downstream application impact would be very helpful.
>
> Thanks for your suggestions. Our method does not require any prior knowledge and pre-processing, thus it can be applied to any other tasks in SE(3) space like point cloud. Actually, the generalization of this method is what we plan to do in the next step, we have tried a toy example to generate simple shapes such as cubes without any other input, and it indicates that this method can be generalized to other scenarios.

---

> ### Comment · Reviewer_K1M9 · 2023-11-21
>
> I appreciate the reviewers' response and agree that reframing the paper to focus and motivate a new kernel for modeling distance distribution would greatly strengthen the paper, especially if the authors can include more experiments and results in other problem settings since these issues are also brought up in other reviewers' feedback.
>
> However, I cannot change my rating as the paper currently stands. I hope reviewers' feedback here can help the author iterate further.

---

### Author Response · Authors · 2023-11-19

Thanks for the thoughtful feedback from all reviewers.  In this response, we address **the topic of Torsional Diffusion**:

We have comparatively analyzed SDDiff and Torsional Diffusion in Appdix. F. We strongly believe that the reason why torsional diffusion surpasses most of the pure data-driven models is that using RDkit provides strong prior in generating the local structures. Some methods [1] use RDkit to first generate conformations and then cluster the results. Such a simple algorithm is already comparable with the torsional diffusion, as shown in the following table.
|                           |  GEOM-QM9  |        |         |        | GEOM-Drugs |        |         |        |
|:-------------------------:|:----------:|:------:|:-------:|:------:|:----------:|:------:|:-------:|:------:|
|                     | COV(%)↑  |        | MAT(Å)↓ |        |   COV(%)↑  |        | MAT(Å)↓ |        |
|             Method              |    Mean    | Median |   Mean  | Median |    Mean    | Median |   Mean  | Median |
| rdkit+torsional diffusion |    92.8    |  100   |  0.178  |  0.147 |    55.2    |  56.9  |  0.778  |  0.729 |
|      rdkit+clustring      |    97.3    |   100  |  0.189  |  0.177 |    48.4    |  44.3  |  0.806  |  0.782 |

---

> ### Comment · Reviewer_XVGd · 2023-11-20
>
> Could the authors explain what the TD and clustering comparison is trying to explain in relation to SDDiff? What is the issue with using RDkit as a strong prior? If the task was molecular design, then using RDkit to initialize would be cheating but that is not this. I am not convinced TD and SDDiff cannot be compared quantitatively on this task.
>
> Where are these numbers from? These rdkit+clustering numbers do not match what's reported in Table 1 of Zhou et al. For TD, why are the recall numbers used for QM9 while the precision numbers are used for Drugs?

---

> > ### Author Response · Authors · 2023-11-21
> >
> > First of all, though the task is conformation generation, we emphasize again that our aim is ''a better prior-free modeling'' rather than ''pushing the performance margin''. We believe ICLR is a top conference that can accommodate a diversity of topics. A **comparison** is necessary when it can support our claim, not when it does not support what we don't claim.
> >
> > We respond to your concerns as follows.
> >
> > 1. As shown in [1], a learning-free RDkit+clustering scheme can achieve very similar performance on QM9. This in turn demonstrates the ''learning'' part in TD does not contribute much to the final performance. Rather, the prior knowledge integrated in RDkit contributes the most. As we focus on learning-based generation without any prior knowledge, directly comparing SDDiff and TD is inappropriate. But you advised, we still think more discussion on TD can be helpful, and will revise our final version accordingly.
> >
> > 2. In terms of the numbers in the table. We use the same scoring threshold as the Torsional Diffusion. That is, on the QM9, $\delta = 0.5$, on the Drugs, $\delta = 0.75$. In the original Table 1 of Zhou et al, for the Drugs dataset, they used $\delta = 1.25$.
> >
> > [1] Zhou, G., Gao, Z., Wei, Z., Zheng, H., \& Ke, G. (2023). Do Deep Learning Methods Really Perform Better in Molecular Conformation Generation?. arXiv preprint arXiv:2302.07061.

---

### Comment · Area_Chair_VR5Z · 2023-11-20
**To all reviewers: please respond to the authors' rebuttal**

Dear reviewers,

The window for interacting with authors on their rebuttal is closing on Wednesday (Nov 21st). Please respond to the authors' rebuttal as soon as possible, so that you can discuss any agreements or disagreements. Please acknowledge that you have read the authors' comments, and explain why their rebuttal does or does not change your opinion and score.

Many thanks,

Your AC

---

### Meta-Review · Area_Chair_VR5Z · 2023-12-11

**Metareview:**

The reviews for this paper were unanimous in their recommendation to not accept this paper, and unfortunately the rebuttal phase did not change the reviewers' perspective. Among other things, concerns were raised on insufficient empirical evidence to demonstrate convincingly the use of the proposed method for conformer generation. I hope that the reviews are helpful for the authors to improve their manuscript, and in particular that the authors can improve their work by including demonstrations that their method can be applied more broadly beyond conformer generation, as was discussed during the rebuttal phase. The recommendation is to reject this paper. I would like to thank the reviewers for their reviews.

**Justification For Why Not Higher Score:**

All reviewers are against accepting this paper.

**Justification For Why Not Lower Score:**

n/a

---

### Decision · Program_Chairs · 2024-01-16

Reject